# Effects of Trace Metals and Municipal Wastewater on the Ephemeroptera, Plecoptera, and Trichoptera of a Stream Community

**DOI:** 10.3390/biology11050648

**Published:** 2022-04-24

**Authors:** Marek Let, Jan Černý, Petra Nováková, Filip Ložek, Martin Bláha

**Affiliations:** Faculty of Fisheries and Protection of Waters, South Bohemian Research Center of Aquaculture and Biodiversity of Hydrocenoses, University of South Bohemia in České Budějovice, Zátiší 728/II, 389 25 Vodňany, Czech Republic; cernyj18@frov.jcu.cz (J.Č.); novakovapetra@frov.jcu.cz (P.N.); lozekf@frov.jcu.cz (F.L.)

**Keywords:** anthropogenic disturbances, aquatic insect, environmental gradients, heavy metals, industrial pollution, wastewater treatment plant

## Abstract

**Simple Summary:**

Mayflies (Ephemeroptera), stoneflies (Plecoptera), and caddisflies (Trichoptera) (EPT) are aquatic insects that are well known to the general public and are commonly used as indicators of environmental quality in water management. Knowledge of how EPT communities react to human-induced gradients in real environments can be important, for example, during the assessment of the implications of newly planned or currently active human disturbances for natural or cultural landscapes. We sampled a stream ecosystem affected by mining and smelting industries and communal wastewaters with pronounced concentrations of cadmium, lead, and zinc, as well as high levels of pesticides, pharmaceuticals, illegal drugs, and sewage-derived organic matter. Changes in other environmental factors such as increases in temperature were also studied at the affected sites. The abundance and species richness of stoneflies fell rapidly at the study sites. The richness of mayfly families also declined, from four to one, even though overall mayfly abundance was not affected. Conversely, the abundances of caddisflies were higher at the affected sites, and their richness did not decrease. This study will provide feedback for ecotoxicologists who perform better controlled and manipulated tests in laboratories, although any such test results are limited by simple artificial environments.

**Abstract:**

Abundances of EPT larvae sampled in a Central European locality affected by mining and smelting, as well as by the continual inflow of treated communal wastewaters (WWs), were recorded. High concentrations of trace metals in water (maximum 1200 µg·L^–1^ for zinc) and sediments (maximum 140,000 mg·kg^–1^ in dry weight for lead) were found at the most contaminated sites. The highest loads of pesticides, pharmaceuticals, and illegal drugs were found under the WW effluent. Other associated factors such as the physicochemical parameters of the water and alterations to microhabitats were also evaluated and taken into account. Although EPT richness was lower at affected sites, abundances did not fall. Stoneflies were dominant at unaffected sites, while caddisflies dominated at affected sites. Only baetid mayflies were detected at the sites contaminated by trace metals and WWs; ephemerellid, heptageniid, and leptophlebiid mayflies were absent from these sites. The site contaminated by trace metals was also inhabited by numerous limnephilid caddisflies, in which limb malformations were detected in up to 11.8% of all specimens of a single taxon. Downstream from the entrance of the WWs, the locality was dominated by hydropsychid caddisflies. The increasing prevalence of predator or passive filter-feeding strategies in these EPT communities was significantly related to increasing water conductivity and acute ecosystemic exposure to ‘poorly treated’ WWs.

## 1. Introduction

The aquatic insect orders of mayflies (Ephemeroptera), stoneflies (Plecoptera), and caddisflies (Trichoptera) (henceforth, EPT) are frequently used indicators of environmental quality, particularly in running waters [1,2]. Their larvae dominate in unpolluted headwaters where proportions of fine sediment deposits are low and are essential for correct nutrient flow cycling in these environments [3,4,5]. The contribution of EPT taxa adapted to conditions in downstream ecosystems (lower rhithral and potamal zones) is likewise important, although populations of these specialists, namely certain species of mayflies and stoneflies, are prone to be decimated by the cumulative effects of intensive human activities in lowland areas [6,7,8].

Given their rapid expansion through the natural environment, the ecological impact of human-induced gradients needs to be monitored, and possible threats are objectively predicted. The contamination of stream ecosystems by toxicants is a global issue; nevertheless, engineering work severely modifies the morphology of whole catchment areas and causes substantial changes in hydrological regimes, habitat structure, and water quality [8,9,10]. Hence, communities in running waters in cultural landscapes are habitually subjected to multiple anthropogenic factors [11]. Therefore, the manipulative testing of stressors within controlled conditions is now seen as increasingly important. Despite this, natural experiments (observational studies) are still needed to reflect the validity of derived results and vice versa [12].

The experiment described in this work explored an EPT community in a temperate zone in the European Central Highlands ecoregion [13], specifically, in a catchment area severely affected by industrial pollution (trace metals) and treated municipal wastewaters (WWs) whose levels are relatively high in terms of current European standards [14,15]. Measured confounding environmental variables (EVs)—concentrations of pesticides, physicochemical parameters of water, choriotopic composition, and current velocity—were included in the analyses of the changes in community composition. The shift in artificially/hierarchically assigned values for feeding strategies in the EPT community along with measured gradients of the EVs was also tested.

The contamination of streams by trace metals (often generalised as ‘heavy metals’) is a global phenomenon, and many studies tackling the responses of macroinvertebrate communities have identified mining as the cause of this type of pollution [16,17,18,19]. Increased concentrations of cadmium, copper, lead, and zinc—the so-called ‘bivalent metals’ that bind to both sulphur/nitrogen and oxygen groups—are most often reported as the cause of shifts in stream macroinvertebrate compositions in real environments or as the potential origin of impaired bottom-up control in ecosystems [20,21]. It is easy to misinterpret an oversimple comparison of concentrations measured in a real environment with the lethal values estimated for several EPT taxa under controlled conditions since many contradictions between observational studies and laboratory or mesocosm experiments exist in assessments of the toxicity of trace metals for aquatic insects [22].

Effluent from wastewater treatment plants (WWTPs) often represents an ‘entrance gate’ for many pollutants, including sewage-derived particulate organic matter (SDPOM), nutrients, pesticides, active pharmaceutical compounds (PhACs), synthetic organic compounds, phthalates, and trace metals into streams [14,23,24,25,26,27]. Over the past two decades, there has been a general decrease in organic matter and nutrient content in outputs from modernised WWTPs in the European Union [28]. Nonetheless, the continual release of compounds that are resistant to conventional purification technologies from WWTPs still occurs [29,30]. In addition, uncontrolled outflows of raw sewage from WWTPs and sewer infrastructure, especially during storm events, persist and represent a further challenge to the protection of natural water resources [31].

This article attempts to decipher the relationships between these human-induced gradients and EPT taxa and their feeding strategies which could be beneficial and permeable in such an impaired environment. It provides a list of taxa that can be considered tolerant to particular types of pollution, and the possible susceptibility of missing taxa is compared and discussed with previous reports. Since EPT taxa usually represent the essential part of macroinvertebrates in streams [3,4,5] and the shift in feeding strategies may reflect the change in available food resources, the article also provides partial evidence about the human-induced changes in the functioning of lotic ecosystems, namely, in nutrient cycling and energy flow [32].

## 2. Materials and Methods

### 2.1. Locality Description

The two surveyed localities—(i) Obecnický brook and (ii) Litavka river—are situated in central Bohemia (Czech Republic, Central Europe; spring to confluence: (i) N 49.7186939, E 13.8732253–N 49.7083778, E 13.9831206 and (ii) N 49.6563228, E 13.8557881–N 49.9602636, E 14.0847414; Figure 1). EPT larvae were sampled at four sites:(i)Site 1 was a stretch of the Obecnický brook in the Brdy highlands above the Obecnice reservoir (third-order watercourse according to Strahler’s system; N 49.7181961, E 13.9110308–N 49.7176411, E 13.9214808; Figure 1 and Appendix A). This site was assumed to be the least affected by human activities, although forestry management has taken place in the area for many years, and there was once here a military firing range. Although increased acidity in this catchment area due to high emissions of sulphur and nitrogen compounds has been reported in the past, this area seems to have partially recovered [33]. The catchment area consists of spruce plantations, forest-free artificial moorlands (the former artillery range), and peat bogs. The brook’s water has very low turbidity, and its rusty brown colour is likely to be caused by humic substances. By comparison, a few spring-fed tributaries had transparent water (an abundant population of acid-sensitive gammarids was observed in one unnamed tributary). Mature spruces grow along the brook edges, along with a few young alders. Particulate organic matter consists mainly of spruce needles, cones, and twigs, as well as leaf litter from the alders, birches, and old unharvested beeches. The brook bottom is densely covered in places by mosses (e.g., *Fontinalis antipyretica*) and species of Marchantiophyta. Despite not monitoring fish populations by electrofishing, brown trout (*Salmo trutta*) were observed during the macrozoobenthic sampling (Appendix A). Fišer et al. [34] reported the presence of the brook lamprey (*Lampetra planeri*), brook trout (*Salvelinus fontinalis*), and European perch (*Perca fluviatilis*) at this site four years before our study began. Insects were prevalent in the macrozoobenthic samples, although only a few small oligochaetes, crustaceans, and molluscs were found;(ii)Site 2 consists of a stretch of the Obecnický brook below the Obecnice reservoir (third-order watercourse according to Strahler’s system; N 49.7161703, E 13.9312433–N 49.7161931, E 13.9346444; Figure 1 and Appendix A). Discharge here was usually lower than at Site 1 because of continuous water abstraction from the reservoir. Water retention and discharge manipulation also influence the colour and turbidity, which were both different from those observed at Site 1 (from dark brown with greater turbidity to crystal clear). Site 2 is surrounded by forest where little management occurs, and the brook edges are lined by old alders in the lower part, which increases the site’s heterogeneity by forming wide pools or dividing the brook into several branches (the gradient was also approximately one quarter less). Proportionally more leaf litter from deciduous trees was observed here than at Site 1. The brook bottom is covered in places by mosses (*F. antipyretica*). Fish stocks consist of brown trout, brook lamprey, and stone loach (*Barbatula barbatula*) (in decreasing order of abundance; Appendix A). Insects prevailed in the macrozoobenthic samples, although small oligochaetes, crustaceans (*Gammarus* sp.), and molluscs (*Pisidium* sp. and *Ancylus fluviatilis*) were relatively abundant;(iii)Site 3 was on a stretch of the Litavka river below its confluence with the Obecnický brook (fourth-order watercourse according to Strahler’s system; from N 49.7100606, E 13.9884817–N 49.7112883, E 13.9961850; Figure 1 and Appendix A). This site has been heavily affected by the local industrial activity (mining, smelting and processing of silver, iron, lead, zinc, and uranium) that has been active for several centuries [15]. An active industrial complex that recycles, above all, lead waste occupies part of the river’s alluvial plain. Heaps of waste material (especially sodium slag) are still stored several meters from the riverbank. Nevertheless, all mining activity has ceased. Here, the effects of agricultural disturbance are also likely to occur as arable fields cover approximately half of the deforested catchment area (excluding human settlements). The rest of the catchment is covered by pastures and meadows. The effect of the municipal WWs is presumably low because of a relatively small human population upstream. The riparian vegetation has been substantially reduced upstream along the Litavka river (third-order watercourse according to Strahler’s system), which has been channelled in places between artificial banks. In addition, several shallow reservoirs (up to 7 ha), including some small sludge deposits, are situated upstream. We sampled the upper channelled parts with old reinforced banks and little riparian vegetation, as well as a lower unchannelled, naturally heterogenic stretch (morphologically similar to those described for Site 2) with banks lined with old alders and willows (Figure 1). The fish stocks here consist of common minnow (*Phoxinus phoxinus*), brown trout, and European perch (in descending order of abundance; Appendix A). Insects prevailed in the macrozoobenthic samples, and small oligochaetes were relatively abundant, but crustaceans and molluscs were almost totally absent from the macrozoobenthic samples;(iv)Site 4 was a stretch of the Litavka river downstream from its confluence with the Příbramský brook (fourth-order watercourse according to Strahler’s system; from N 49.7113508, E 14.0093011–N 49.7198211, E 14.0133511; Figure 1 and Appendix A). This site was expected to be the most affected by anthropogenic activities because of the combination of industrial pollution from mining and smelting and contamination by treated municipal WWs from the town of Příbram. Treated WWs are continually pumped into the Příbramský brook approximately 900 m upstream from its confluence with the Litavka river. According to Grabicova et al. [14], Site 4 has the greatest concentrations of psychoactive PhACs of all the inspected localities, probably because of the low dilution possibilities (i.e., a small watercourse receiving relatively large amounts of municipal WWs). As at Site 3, we sampled an upper channelled stretch with artificial banks and little riparian vegetation, as well as a lower unchanneled section. Nonetheless, this morphologically heterogenic stretch only had a narrow and irregular fringe of riparian trees (alders and willows), possibly because of the dynamic migration of the river’s course across its alluvial plain [15]. The fish stock consists of common minnow, common roach (*Rutilus rutilus*), chub (*Squalius cephalus*), gudgeon (*Gobio gobio*), brown trout, European perch, common bream (*Abramis brama*), common rudd (*Scardinius erythrophthalmus*), stone loach, and rainbow trout (*Oncorhynchus mykiss*) (in descending order of abundance; Appendix A). The occurrence of limnophilic fish was due to the presence of aquaculture ponds upstream. There was no clear dominance by insects in the macrozoobenthic samples since small oligochaetes, leeches (e.g., *Erpobdella octoculata*, *Helobdella stagnalis*), and the water louse (*Asellus aquaticus*) were very abundant. Molluscs were detected in low numbers.

### 2.2. Sampling and Analysis of Samples

The whole sampling campaign was conducted in 2020, specifically on 11 May, 24 June, 11 August, and 24 September. The macrozoobenthos were sampled using a Surber sampler (30 × 30-cm frame, 500-μm mesh) at four sampling sites and three points (triplicates) at each site (see part 2.1. ‘Locality description’ and Figure 1). Each triplicate consisted of three subplots within a whole plot or site. The location of the whole plot was chosen at random during each sampling campaign. A diversification of samples within each whole plot was made by sampling riffles (shallow and with high water velocity), inflows into pools (deep and with high water velocity) and the backs of pools (deep and with low water velocity). Samples were processed using a round steel sieve (40 cm diameter, 500 μm mesh) and preserved in 70% technical ethanol. Organisms were sorted in the laboratory from the material taken from the stream bottom. Mayfly, stonefly, and caddisfly larvae were determined to the deepest possible taxonomic level using a binocular microscope and stereomicroscope (Olympus SZX16) and determination keys; their frequencies (abundances) in the samples were recorded. The methodology used to analyse the water samples for pesticides, PhACs and drugs, and the water and sediment samples for trace metals is described in detail in the Appendix A (Section Materials and Methods).

Environmental data were taken for both the whole and subplot levels. The environmental variables (EVs) differing only at the whole-plot level (hereafter referred to as ‘whole-plot EVs’) consisted of (i) physicochemical parameters—conductivity, pH, oxygen concentrations, and water temperature—measured using a HACH^®^ multi-meter, (ii) pesticide and PhAC concentrations in water (measured from samples taken on 11 August 2020), (iii) cadmium, lead, and zinc concentrations measured in sediments and water (measured from samples taken on 24 September 2020), and (iv) daily volumes of sewage (untreated or ‘poorly treated’ municipal WWs) recorded up to 15 days prior to each sampling day at the Příbram WWTP (data provided by 1. SčV JSC, Příbram, Czech Republic). Detailed monthly data of water and sediment analyses from Sites 1, 2, and 3 (2009–2020) and data of actual water discharge from the same sites were provided by the Czech Hydrometeorological Institute (Prague, Czech Republic) and Povodí Vltavy, State Enterprise (Prague, Czech Republic). In addition, the water temperature was continuously monitored during the sampling campaign by four data loggers (TFA) placed approximately in the middle of each site.

The EVs also differed at the subplot level (hereafter referred to as ‘subplot EVs’) in terms of their (i) choriotopic compositions, which were empirically estimated for each Surber sampling spot; (ii) current velocities [m·s^–1^] (minimum, maximum and average), measured continuously for 0.5 min above the Surber sampling spot with a flowmeter (MiniController MC20 with the Flowprobe for MiniWater20, Schiltknecht Messtechnik AG, Gossau, Switzerland) placed close to the bottom, in the water column and close to the water surface; (iii) widths and depths at each subplot; and (iv) classification as riffles, pools, inflows into pools, or backs of pools.

### 2.3. Statistical Analyses

Data were analysed using Canoco 5 version 5.12 (written by Cajo J.F. ter Braak and Petr Šmilauer; Wageningen, the Netherlands, and P. Šmilauer, Czech Republic) [35] and R-studio version 4.1.1 (written by RStudio team; Boston, MA, United States) [36] software. The analysis of shifts in feeding strategies was based on the preferences of individual species according to Graf et al. [37], Buffagni et al. [38], and Graf et al. [39]. The analyses with species community composition as response variables were conducted using unimodal Canonical correspondence analysis (CCA), while analyses with feeding strategies as response variables were performed using linear Redundancy analysis (RDA). Species abundances were log + 1 transformed before the CCA. Detrending was not used because no dependency between positions of samples on the first or second ordination axes was observed. Feeding strategies were averaged by species composition-weighted standardisation (hereafter referred to as ‘averaged feeding strategies’). Only centring by species was applied using RDA.

The significances of axes constrained by (i) site identity, (ii) the identity of sampling times, (iii) whole-plot EVs, and (iv) subplot EVs (see part 2.2. ‘Sampling and sample analysis’) were tested using Monte-Carlo permutation tests (999 permutations used). Permutations were carried out according to the experiment’s hierarchical design: (i) when the whole-plot EVs or the identity of sites were used as explanatory variables, both whole-plots and subplots in the four blocks defined by the four sampling times used as covariates were permuted using cyclic shifts; (ii) when the subplot EVs were used as explanatory variables, only the subplots in the four blocks defined by sampling times were permuted using cyclic shifts; and (iii) when the identity of sampling times was used as an explanatory variable, both whole plots and subplots in the four blocks defined by the four sites were permuted using cyclic shifts. A false discovery rate was used for adjusting *p* values. Forward selection (FS) was used for selecting significant whole- and subplot EVs. ‘Hybrid analysis’ was chosen when tests on visualised constrained axes were not significant (*p* > 0.05). The significance of differences in choriotopic compositions between sites was tested in a similar way; individual values were transformed by the angular (inverse sine) function and the transformed values were used as response variables. This transformation was also applied when the choriotopic compositions were used as explanatory variables.

Generalised linear models (GLMs) were employed to model trends in shifts of multiple averaged feeding strategies along environmental gradients or along the resulting ordination axes. The significance of differences between sites and sampling times in EPT abundances and richness was tested using GLMs with negative binomial and Poisson distributions, respectively (the quasi-likelihood estimation method was applied in the case of the Poisson distributions). The significance of improvements using Generalised linear mixed-effects models (GLMMs) rather than GLMs was tested by likelihood ratio tests. A post hoc test with Tukey’s method for *p* value adjustment was applied.

## 3. Results

### 3.1. Environmental Conditions

The differences between sites are summarised in Table 1. Several environmental gradients, including increasing temperature, pH, and conductivity and decreasing oxygen concentrations, changed along with the longitudinal downstream profile (from Site 1 to Site 4). For more information about the basic physicochemical parameters at other sites in the longitudinal profile, see Appendix A (Section Materials and Methods; Appendix A). The greatest total concentration of cadmium, zinc, and lead in the waters was detected at Site 3, followed by Site 4, while the highest loads of pesticides and PhACs were found at Site 4 below the discharge of effluent from the WWTP (Table 1). Treated WWs constituted 1/13–1/3 of all discharges at Site 4 (Appendix A). The periodical releasing of untreated or ‘poorly treated’ WWs containing organic material and coarse communal waste (mainly hygienic products) also occurred. The monthly released volumes of this sewage ranged from 299 to 130,016 m^3^ in 2020 (Table 2); however, considerable volumes also leaked from sewers upstream from the WWTP but were not included in the analyses. For more information about contamination at other sites along the longitudinal profile, see Appendix A (Section Materials and Methods; Appendix A).

The average current velocities at individual sites were as follows: Site 4 >Site 3 > Site 1 > Site 2 (in descending order from the fastest to the slowest; Table 3). Differences in the choriotopic compositions at sites tested using a RDA were marginal (test on first axis: pseudo-*F* = 1.2, *p* = 0.085; test on all axes: pseudo-*F* = 2.8, *p* = 0.003; Appendix A); only Site 4 and Site 2 were significantly different from the others when tested with the RDA (explained variation = 7.5%, pseudo-*F* = 3.8, *p* (adj.) = 0.025 and explained variation = 6.0%, pseudo-*F* = 2.9, *p* (adj.) = 0.049, respectively). Samples from Site 1 compared with those from Site 2 had a much greater proportion of smaller mineral particles (size < 6.3 cm—‘microlithal’, ‘akal’, and ‘psammal’) and ‘mosses’ (Marchantiophyta only at the Site 1), and a lower proportion of light deposits (‘coarse particulate organic matter’—CPOM and ‘fine particulate organic matter’—FPOM), debris, and xylal (Table 4). Samples from Site 3 resembled most those from Site 1, with similar proportions of smaller mineral particles, deposits, debris, and xylal (Table 4). Finally, Site 4 differed most from the others because of the highest proportions of deposits and debris and the occurrence of filamentous algae and anthropogenic trash (especially wet wipes; Table 4).

### 3.2. EPT Abundance and Richness

In total, 87 EPT taxa were detected in the samples: 11 mayflies, 29 stoneflies, and 47 caddisflies (Appendix A). EPT abundances were not substantially different between sites (Figure 2A; Table 5) or sampling times (Table 5). Nevertheless, total EPT richness estimated for individual sites did show a gradual decreasing and significant trend from Site 1 to Site 4 (Figure 2B; Table 5). Sampling time had a nonsignificant effect on EPT richness (Table 5).

The greatest differences between sites in terms of abundance or richness were observed in stoneflies followed by mayflies (Table 5), both of which were impaired at disturbed Sites 3 and 4 (Figure 2C–E). Conversely, caddisflies were more abundant at disturbed Sites 3 and 4 (Figure 2C), although caddisfly richness and family richness did not significantly differ between sites (Figure 2D,E; Table 5). Significant differences between the four sampling times in abundance or richness were detected in mayflies and stoneflies but not in caddisflies (Table 5). The abundance of mayfly nymphs was lowest in August and highest in May, whilst the opposite was found in stoneflies: the highest stonefly abundance was detected in August and the lowest in May.

The stonefly and caddisfly orders significantly differed in abundance between sites (Table 5). Stonefly abundances rapidly declined at disturbed Sites 3 and 4 compared with Sites 1 and 2, whilst caddisfly abundances were lower at Sites 1 and 2 (Figure 2C). Mayfly abundance did not significantly differ between sites (Figure 2C, Table 5) but marginally differed between sampling times (Table 5). No significant interaction was detected between the four-level factor site and sampling time.

The total abundances and species richness shown in Figure 2F,G indicate that practically only members of the Baetidae family (namely *Baetis fuscatus*, *B. rhodani*, *B.* aff. *scambus*, and *B. vernus*) and the Leuctridae family (stoneflies) were detected at disturbed Sites 3 and 4. At these sites, *Habrophlebia lauta* (Leptophlebidae) and *Seratella ignita* (Ephemerellidae) all but completely disappeared, while *Paraleptophlebia submarginata* (Leptophlebiidae), *Ecdyonurus torrentis*, and members of the genus *Rhithrogena* (both Heptageniidae) were totally absent, although these latter taxa were detected upstream at the interconnected Sites 1 and 3 (Figure 2F, Appendix A). Of the stoneflies, only *Leuctra albida*, *L. fusca*, and *L. geniculata* were detected at Site 3 (*L. geniculata* was only detected at this site, Appendix A) and only *L. fusca* in low abundances and body sizes was detected at Site 4 (Figure 2G, Appendix A). *Siphonoperla torrentium* (Chloroperlidae), several species belonging to the genera *Protonemura*, *Amphinemura*, and *Nemoura* (Nemouridae), all very abundant at Sites 1 and 2, along with *Diura bicaudata* and *Perlodes* aff. *microcephalus* (Perlodidae) were completely absent in the samples taken from disturbed Sites 3 and 4 (Figure 2G, Appendix A). Nevertheless, compared with Site 1, the richness of Leuctridae decreased also at Site 2 below the water reservoir (Figure 2G), and, for instance, *Leuctra nigra*, frequent in samples from Site 1, was missing from samples from Site 2 (Appendix A).

Of the caddisflies, Site 1 had a greater richness of trichopterans than Site 2; the abundant family Philopotamidae, for example, was no more detected at Sites 2, 3, and 4; finally, members of the Goeridae were detected only sporadically at particular sites (Figure 2H). On the other hand, the richness of Hydropsychidae was lowest at Site 1. *Odontocerum albicorne* (Odontoceridae) and *Sericostoma flavicorne*/*personatum* (Sericostomatidae), very abundant at both Sites 1 and 2, were completely missing from the samples from Sites 3 and 4 (Figure 2H, Appendix A). One of the dominant families at Site 3 was Limnephilidae, represented above all by species from the genera *Potamophylax*, *Halesus*, and *Chaetopteryx* (in descending order of total abundance; Figure 2H, Appendix A). One of the dominant families at Site 4 below the WWTP was Hydropsychidae, represented by *Hydropsyche siltalai*, *H. angustipennis*, and *H. incognita*/*pellucidula* (in descending order of total abundance; Figure 2H, Appendix A); the population of limnephilids at Site 4 was substantially lower (Figure 2H). At both disturbed Sites 3 and 4, the family Polycentropodidae was much more abundant than at Sites 1 and 2 (Figure 2H). A substantially greater proportion in the total abundances of *Cyrnus trimaculatus* than *Polycentropus flavomaculatus* was observed at Site 4 than at Site 3 (2.54 and 0.03, respectively; Appendix A). The Leptoceridae represented by *Athripsodes bilineatus* was most abundant at Site 3, while the Psychomyiidae, represented only by *Psychomyia pusilla*, were detected only at Site 4 (Figure 2H; Appendix A). The total abundances of the family Rhyacophilidae were almost identical at all sites (Figure 2H). Even though the clear morphological determination of larvae from the group *Rhyacophila* sensu stricto is not always possible [40], *Rhyacophila* cf. *aurata* was dominant in samples from Sites 3 and 4 (especially Site 3), whilst *Rhyacophila nubila* gr. was commonest in samples from Sites 1 and 2.

### 3.3. Shift in EPT Community Composition along Environmental Gradients

The differences in species compositions between sites were significant (test on first axis: pseudo-*F* = 1.9, *p* = 0.001; test on all axes: pseudo-*F* = 4.0, *p* = 0.001; Appendix A), as were the differences in composition of species between sampling times (but only the composition of specimens > 0.5 mm in size; test on first axis: pseudo-*F* = 0.7, *p* = 0.024; test on all axes: pseudo-*F* = 1.8, *p* = 0.001; Appendix A). The whole- and subplot EVs significantly correlated with the shift in species composition (*p* < 0.05); those selected by FS are shown in Figure 3 and Figure 4, respectively.

### 3.4. Shift in Averaged Feeding Strategies along Environmental Gradients

The differences in compositions of the averaged feeding strategies between sites were significant in general; however, only the composition at Site 4 was significantly different from all other sites (*p* < 0.05; Figure 5A). There was also a significant shift in the whole-plot EVs (*p* < 0.05). Significant relationships with ‘conductivity’ (32.5% of explained variability), ‘total volume of released sewage for three days prior to sampling time’ (5.9% of explained variability), ‘oxygen concentration’ (2.8% of explained variability), and ‘total concentration of PhACs’ (2.3% of explained variability) were detected (*p* < 0.05). ‘Passive filter feeder’ and ‘predator’ feeding strategies were dominant in the WW-polluted environment; the ‘grazer and scraper’ feeding strategy seemed to be limited by the most polluted WW; and the ‘gatherer/collector’ and ‘shredders’ feeding strategies were generally suppressed in the most polluted environment (Figure 5B and Appendix A). The shift in the subplot EVs was not significant (test on first axis: pseudo-*F* = 0.6, *p* = 0.379; test on all axes: pseudo-*F* = 1.9, *p* = 0.168).

### 3.5. Malformations and Mortality of Caddisflies

A substantial proportion of final instar caddisfly larvae in *Halesus* cf. *tesselatus*, *Potampohylax* aff. *rotundipennis*, and *R.* cf. *aurata* sampled at Site 3 had malformed limbs (11.8%, 7.0%, and 7.7%, respectively). Peculiarly fused terminal parts of walking legs (between tibiae and tarsal claws) or overall shortened walking legs were observed in the two given limnephilid taxa; deformed and shortened anal claws were observed in *R.* cf. *aurata* (Figure 6). Malformed specimens of *H.* cf. *tesselatus* were also observed at Site 4 (28.6%), although at the latter site, the total number of sampled specimens was substantially lower than at Site 3 (7 vs. 76 specimens at Site 3). There were no malformed specimens in the abundant *R.* cf. *aurata* in Site 4; however, there was one malformed specimen out of a total of 20 specimens of *R. nubila* gr. at Site 4.

We also observed a drift of dead limnephilid pupae at Site 3 in September during the macrozoobenthic sampling. We found dead and decaying larvae and pupae of *Drusus annulatus*, *H.* cf. *tesselatus*, and Limnephilidae gen. sp. inc. in the given samples (14 dead limnephilid specimens and 7 living limnephilid specimens in total). Several dead and decaying pupae of *Hydropsyche* sp. were also found in the samples from Site 2 taken in June.

## 4. Discussion

### 4.1. Variability in the EPT Community along an Environmental Gradient

The EPT community can significantly reflect the quality of the environment (Figure 3, Figure 4 and Figure 5). Even though our study only consisted of a small-scale natural experiment (i.e., we only sampled two mutually interlinked watercourses), the results match, in general, those of a study of whole macroinvertebrate communities carried out in 23 Swiss streams [41]. As in our work, in the Swiss study, about 30% of community variability was explained by water quality; conductivity (which correlates to total dissolved solids) together with oxygen concentrations explained 10.1% in our study, while concentrations of pesticides explained 3.0% of community variability in both studies (Figure 3).

The changes in EPT abundance and richness related to pollution by trace metals observed in our study generally resemble reports from other sites, the greatest similarity being in the susceptibility of mayflies—and especially heptageniids—to this type of pollution (Figure 2). Sites 3 and 4 were both warmer—maximums up to 23 °C—than unpolluted Sites 1 and 2 (Table 1; Appendix A), a relevant finding given that the temperature gradient, which is often related to the toxicity of chemical compounds and the solubility of oxygen in water, is considered a good predictor of change in the composition of EPT communities [42]. The increasing temperature gradient observed when moving downstream is undoubtedly natural; however, the difference in the temperature regime between the contaminated and uncontaminated sites is also likely to be the product of the canalisation, the removal of the riparian vegetation, and the presence of shallow reservoirs upstream on the Litavka river (see part 2.1 ‘Locality description’ [43]). That many species of mayflies and stoneflies would avoid high temperatures, mine effluent and certain trace metals was predicted beforehand [17,18,19,44,45,46]. Our results show that baetids were the most resistant (or resilient) to the trace metal contamination, and within this group, the dominant taxa were the *Baetis fuscatus* group (i.e., *B. fuscatus* and *B.* aff. *scambus,* whose separation was not possible since the taxon identified and presented here as *B.* aff. *scambus* shared features of both species). The total abundance of *B. fuscatus* gr. compensated for the decrease in abundances of other mayfly taxa so that this small-sized baetid could act as a proxy for other mayflies in human-disturbed sites. The other most commonly detected baetids, *B. rhodani* and *B. vernus*, did not show any substantial changes in abundance between sites. Nevertheless, trace metal pollution can have an effect later on in the life cycle of baetids and other aquatic insects by rapidly causing increased mortality in emerging imagines (or subimagines) [47,48].

Stoneflies are generally the aquatic insects most sensitive to conditions modified by humans [49,50]. There was significantly decreased richness in stoneflies—even at Site 2 compared with Site 1—and their abundance only significantly fell at Sites 3 and 4 (Figure 2; Table 5). We interpret their decrease in richness at Site 2, situated below the reservoir, as the result of the discontinuity in the river network, i.e., a change in the deposition regime (a lower proportion of smaller mineral particles and a greater proportion of light deposits; Table 4), and the separation of Site 2 from the network of tributaries in the crenal zone situated upstream from the reservoir (Figure 1). However, there are several indications that stoneflies are less sensitive to metals than mayflies. Many stoneflies (as well as caddisflies) are more tolerant to increased acidity than many mayflies [51]. Lower pH leads to natural access to metals (e.g., aluminium) originally bound to substrates [52,53]. Observational work studying the effects of contamination from mining has reported relatively high resistance in some stoneflies, including *Amphinemura sulcicollis*, *Leuctra fusca*, and *L. inermis*, taxa that are similar to those found in our study [16,19]. Little evidence exists regarding the levels of trace metals concentrations that are lethal for stoneflies [54], although certain North American stoneflies have been found to withstand (in significantly reduced numbers) concentrations of 11, 120, and 1100 µg·L^–1^ of cadmium, copper, and zinc, respectively, for 10 days in microcosm conditions. By comparison, numbers of heptageniids and *Baetis tricaudatus* fell significantly in only half of these concentrations [46].

Therefore, the significant reduction in stonefly abundance and richness at both contaminated sites cannot be conclusively attributed to pollution by trace metals. The absence there of most of the stoneflies detected upstream could also be due to water warming since most stoneflies prefer lower temperatures than those recorded at Sites 3 and 4 [39,42]. However, the interaction of multiple effects, including habitat degradation and several types of pollution, are assumed to take place.

According to our data, the less sensitive stoneflies were able to withstand the heavy industrial trace metal pollution but were then eliminated by the municipal WW, despite the advanced technology used in the WW treatment [14]. A synergic effect of multiple stressors is likely to occur. We can demonstrate that specimens of all sizes of *Leuctra fusca* gr. (species identified as *L. albida* and *L. fusca* in this study [55]) and *L. geniculata* occurred in the water contaminated by trace metals at Site 3 (Appendix A) but not in the water contaminated by municipal WWs. Only *L. fusca* gr. (the only species identified as *L. fusca*) of small sizes were detected at Site 4, contaminated by both trace metals and municipal WWs. It is important to note that the organic pollution produced by the WWs sometimes reaches high levels. According to unpublished data provided by the Czech Hydrometeorological Institute and Povodí Vltavy, State Enterprise, the average estimated five-day biochemical oxygen demand (BOD_5_) was equal to 2.39 mg·L^–1^ at Site 4; however, the estimated maximum was 17.00 mg·L^–1^ just four months before the beginning of our sampling campaign. In the Příbramský brook (Auxiliary Site 4 downstream from the effluent from the WWTP, see Appendix A), the estimated BOD_5_ at the same time was 24.00 mg·L^–1^ (the average and maximum of the estimated BOD_5_ were 4.76 and 60.00 mg·L^–1^ in 2013–2020, respectively).

Caddisflies were the least harmed insects in the EPT community. Their richness was not significantly different between sites, and their abundance was significantly higher at Site 4 than at Sites 1 and 2 (Figure 2; Table 5). Many caddisflies are acid-tolerant or acid environment specialists (e.g., *Chaetopterygopsis maclachlani* and *Rhyacophila polonica* detected during our study [51,56]), and some are reported to be tolerant to trace metal pollution (e.g., *Hydropsyche* sp., leptocerids, *Rhyacophila* sp. [17,18,46]). Ubiquitous caddisfly taxa can occupy warmer sites polluted by organic materials with low oxygen concentrations (e.g., potamal hydropsychids and *Psychomyia pusilla* [56,57]). Nevertheless, caddisflies are generally very sensitive to pesticides and avoid sedimentation and droughts, as does the EPT group in general [11,58,59].

The vast majority of EPT taxa situated on the right-hand side of the CCA biplot (Figure 3A) were caddisflies. The relative abundances of these caddisflies were positively correlated to the human-induced gradients. The caddisfly communities at Sites 1 and 2 were quite similar; however, net-spinning philopotamids were lacking from Site 2 and were replaced by net-spinning hydropsychids and polycentropodids (Figure 2H). Changing conditions along the longitudinal profile (e.g., food source and current velocity) probably played an important role in shaping the net-spinning caddisfly community. Philopotamids prefer diatoms and fine detritus and have been reported to be quite sensitive to organic pollution [60,61]). According to our data, philopotamids were found at the subplots with the greatest maximum water velocity (0.73–0.97 m·s^–1^). Slower water speeds caused by water abstraction [60] combined with the natural longitudinal stream gradient [37,60] favoured hydropsychids and polycentropodids. Unlike philopotamids, hydropsychids and polycentropodids are either facultatively or obligatory carnivorous [37,62].

The altered conditions at Sites 3 and 4 probably prevent the occurrence of the eruciform caddisflies *O. albicorne* and *S. flavicorne*/*personatum*, which were dominant in the samples from Sites 1 and 2. The warmer temperatures (and poorer oxygen conditions) are expected to have a negative effect. Haidekker and Hering [42] have reported that *O. albicorne* tolerates well mean summer temperatures up to 16 °C. However, this figure was exceeded at Site 4 by 1.24 °C (Appendix A), and so, based on individual measurements (Table 1), we assume that a similar temperature regime exists at Site 3. Furthermore, both these taxa are partially endobenthic and are more exposed to the toxic compounds that accumulate in sedimented particles [63]; the long larval development time in *S. flavicorne*/*personatum* [64] will also prolong the exposure of its larvae to toxicants. On the other hand, high abundances of large eruciform caddisflies—limnephilids of the genus *Halesus* and *Potamophylax*—were found at Site 3. These taxa are probably tolerant to trace metal pollution. Nonetheless, we found malformed walking legs in up to 11.8% of individuals that, in light of previous studies, could be a sign of trace metal contamination [65] or other persistent and hazardous compounds generated by the industrial complex upstream from Site 3 [66].

The lower abundances and richness of limnephilids observed at Site 4 downstream from the WWTP effluent could be a consequence of the interaction between trace metals and WW pollution (or the WW pollution alone). Hydropsychids, polycentropodids, rhyacophilids, and psychomyiids (*P. pusilla*) were most abundant downstream from the WW effluent. The taxa are shown in the upper right-hand corner of Figure 3A were positively related to the ‘total volume of released sewage three days prior to the sampling’ (hereafter referred to as ‘3-day sewage volumes’), which probably contained a high proportion of SDPOM—a mixture of organic detritus and microorganisms such as bacteria and algae that is a high-quality food resource downstream from WW effluents [23]. On the other hand, the released sewage could trigger a drift of benthic prey organisms with greater oxygen demands (e.g., mayfly nymphs), and SDPOM could be used by periphyton and macrophytes (aquatic and riparian), which can serve as microhabitats or food. For example, the carnivorous *C. trimaculatus* with a more pronounced microhabitat preference for macrophytes [37] was more common than *P. flavomaculatus* at Site 4 than Site 3. The most abundant hydropsychids can use multiple food resources and, aside from predating on small benthic organisms and consuming detritus, they also graze on algae [37,67]. Besides crustaceans, insects (including EPT), and terrestrial prey, detritus also constitute a substantial portion of gut content in polycentropodids [68]. It cannot be rejected that hydropsychids and polycentropodids could have a detrimental impact on the populations of other EPT taxa. Nevertheless, their net spinning increases heterogeneity of current velocities and nets represent valuable microhabitats, e.g., positive relationships to ephemerellid mayfly and chironomids were reported [69,70]. We associate the high abundance of early instars of *H. siltalai* in September at Sites 3 and 4 with the presence of aquatic mosses (Figure 4).

The proportion of mosses in the choriotope (explained community variation = 5.3%) and mean current velocity (explained variation community variation = 5.2%) had the highest parsimonious and significant power to explain EPT community composition at the subplot level (Figure 4). These two variables were positively correlated because mosses were detected mainly in shallow, fast-flowing habitats. Mosses (as well as roots) provide better vertical zonation possibilities for aquatic organisms, as well as food and shelter [71]. The information provided by Buffagni et al. [38], Graf et al. [37], and Graf et al. [39] does not correspond particularly well to our results: of the species best corresponding to the presence of aquatic mosses only *Hydropsyche saxonica, Plectrocnemia conspersa*, and *Protonemura hrabei* were reported to have a certain preference for macrophytes (mosses included). Nevertheless, Krno [72] suggests that the goerid *Lithax niger* is a bryophyte dweller and reports that *Rhyacophila obliterata* can only live on emergent mosses, whilst *R. nubila*, *R. polonica* can also live on submerged mosses. Similar findings have been reported by Glime [71]. Conversely, Graf et al. [37] consider all goerids and rhyacophilids to inhabit micro-/mesolithal and macro-/megalithal and suggest that they are ‘specialised for lithal’.

A high mean velocity best corresponded to the highest relative abundance of *H. pellucidula*/*incognita* (the larvae of these two species are often not separable [40]): fitted optimum over 0.60 m·s^–1^; *R.* cf. *aurata* and *R. nubila* gr.: fitted optimum around 0.50 m·s^–1^; and *H. siltalai*: fitted optimum 0.47 m·s^–1^ (Figure 4). The mayfly *B. fuscatus* had a fitted optimum around 0.35 m·s^–1^ (Figure 4). By contrast, according to our data, *O. albicorne* had a fitted optimal mean current velocity around 0.14 m·s^–1^, *S. flavicorne*/*personatum* 0.10 m·s^–1^, and the mayfly *H. lauta* less than 0.10 m·s^–1^.

### 4.2. Shift in Averaged Feeding Strategies in the EPT Community along Environmental Gradients

The analysis of shifts in ecological traits and, above all, their relationship to environmental characteristics has recently become a key topic in community ecology [73]. Substantial changes in the composition of feeding strategies (functional feeding groups) due to disturbances are assumed to occur in the macrozoobenthos [74,75]. Significant differences in the composition of average EPT feeding strategies were detected between localities (Figure 5A). Conductivity and 3-day sewage volumes were found to be the most parsimonious significant predictors in our dataset (Figure 5B). Both these EVs were positively related to the input of various materials and were positively related to both passive filter feeder and predator feeding strategies (represented almost only by caddisflies; Appendix A). It suggests in higher susceptibility of EPT shredders to the pollution or unavailability of the CPOM (main food resource utilised by shredders) edible for these organisms (e.g., crystalline deposits containing trace metals can precipitate on the external surface of alder leaves because of fungal grown [21]). Even though shredder abundance will be compensated by numerous *Asellus aquaticus* and gatherers/collectors by chironomids and oligochaetes (see part 2.1. ‘Locality description’), the abundance of winged adults of the aquatic insect directly utilising CPOM may be substantially lowered and could cause an alteration in nutrient and energy flows through the environment.

Despite that, 3-day sewage volumes positively correlated to the responses of passive filter feeders, predators, and shredders and negatively to the responses of gatherers/collectors (Appendix A), the greatest 15-day sewage volumes (Appendix A) positively correlated to the responses of gatherers/collectors and grazers and scrapers. Nevertheless, this trend was not strongly associated with the ‘15-day sewage volumes’ because the dominance of gatherers/collectors and grazers and scrapers was caused by a high abundance of nymphs of the mayfly *B. fuscatus* gr., which probably colonised this site after drifting from an upstream stretch after a high discharge event (up to 8.5 times the average discharge). This hydrological situation occurred at the same time as the release of a high volume of sewage. The sewage release ended four days before the sampling time. The greatest similarity in the overall EPT community of samples to the samples from Site 3 is shown in the overlap of the ellipses in Appendix A. This similarity was subsequentially lost when the heavy rains stopped: the greatest dissimilarity between samples from individual sites was observed in August, followed by September. Nonzero values of high 3-day sewage volumes were linked to EPT compositions sampled in both these months, as well as in May. The significant linkage between 3-day sewage volumes and the composition of feeding strategies, as a result of FS in RDA (Figure 5B), supports the presumption that the population of campodeiform net-spinning caddisflies is directly or indirectly dependent on the supply of ‘poorly treated’ WWs.

## 5. Conclusions

This study highlights the impacts on the EPT community of trace metal pollution originating from mining and smelting and effluent from a municipal WWTP, which releases continuously treated WW and, periodical, ‘poorly’ treated WW. According to our data and previously published studies, trace metal pollution can have quite different consequences for the EPT community than, for example, pesticides and several caddisflies, especially limnephilids, were found to be very abundant at the affected site. Additionally, malformations in caddisfly larvae and mortality of caddisfly pupae were observed. Even though the affected sites were warmer, the absence of ephemerellid, heptageniid, and leptophlebiid mayflies is attributable to trace metal pollution, given the results from previous studies. The factors responsible for the decline in the stonefly community are less clear, and, except for higher water temperature, other kinds of pollutants and habitat degradation can be assumed to have a negative impact. Nevertheless, *L. fusca* gr. and *L. geniculata* can be considered tolerant to trace metal pollution but sensitive to WW pollution or its combination with trace metals. The municipal WW combined with trace metal contamination led to the greatest changes in the EPT community, including changes in the composition of feeding strategies. Caddisfly passive filter feeders and predators—mainly hydropsychids, polycentropodids, and rhyacophilids—dominated the most polluted environment. EPT shredders and collectors/gatherers dominated in unaffected sites. Our results demonstrate how the EPT community reacts to human-induced gradients in natural environments, which is important knowledge for assessing the implication of planned or current human disturbances in natural or cultural landscapes.

## Figures and Tables

**Figure 1 biology-11-00648-f001:**
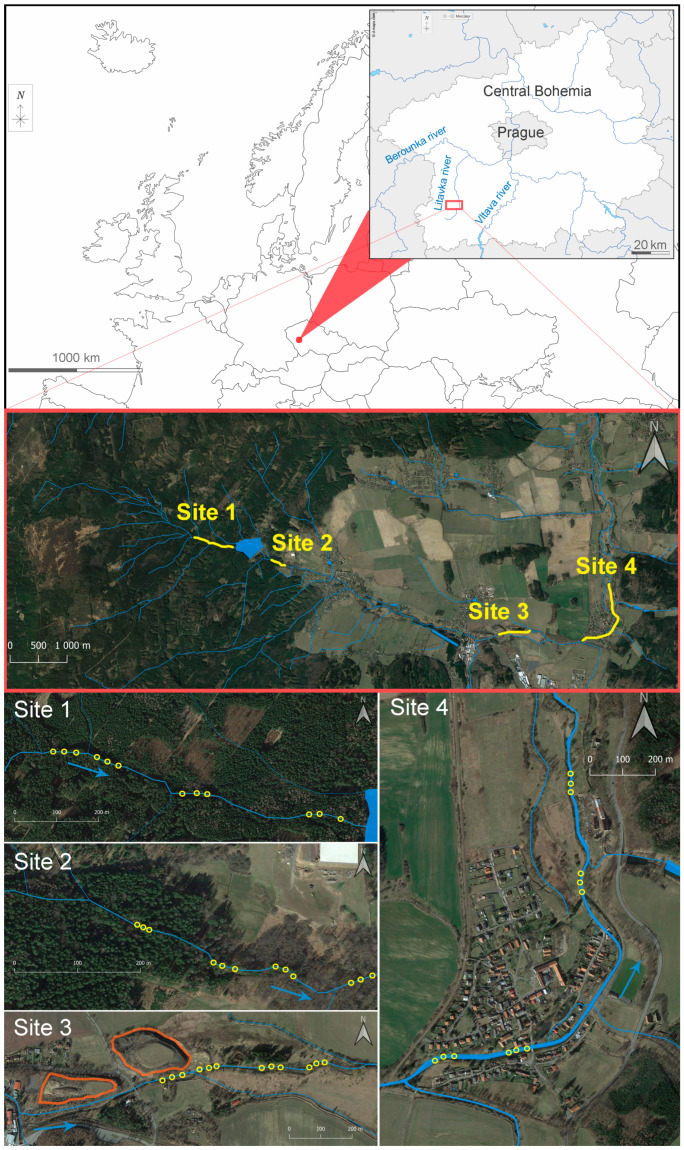
Map of the study area. The yellow circles indicate the sampling points. A set of three sampling points represents a whole plot. The contoured orange-coloured areas in the close-up view of Site 3 (lower left-hand corner) are slag heaps and waste material generated by the smelting industry. The river’s flow direction is indicated by blue arrows.

**Figure 2 biology-11-00648-f002:**
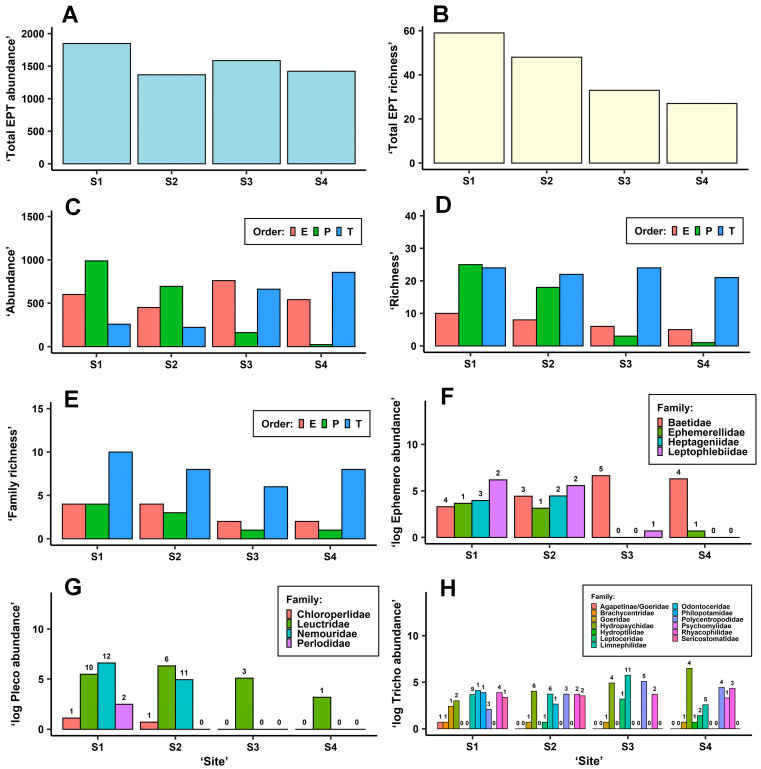
Bar graphs visualising (**A**,**C**) ‘abundance’ (number of individuals), (**B**,**D**) ‘richness’ (number of taxa), (**E**) ‘family richness’ (number of families), and (**F**–**H**) abundances and richness of individual detected families of Ephemeroptera, Plecoptera, and Trichoptera, respectively, summarised for each ‘site’ from the samples taken at all sampling times. Values in the bar graphs (**F**–**H**) are transformed by a natural logarithm, and the ‘richness’ is displayed above each column. E—Ephemeroptera, P—Plecoptera, T—Trichoptera.

**Figure 3 biology-11-00648-f003:**
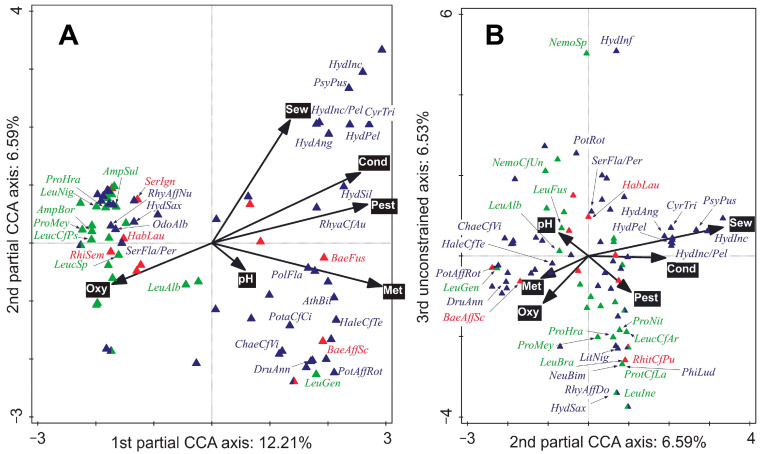
Species + whole-plot environmental variables (EVs) biplots: the first 33 best fitting species are labelled (Ephemeroptera, Plecoptera, and Trichoptera labelled in red, green, and blue, respectively). A partial canonical correspondence analysis (CCA) was used: (**A**) first and second ordination axes; (**B**) second and third ordination axes. Ordination axes were constrained by the EVs selected by forward selection, which accounted for 30.48% of the explained variation. Test on first axis: pseudo-*F* = 0.5, *p* = 0.001; test on all axes: pseudo-*F* = 2.3, *p* = 0.001. Conditional effects (from the greatest to the lowest explained variation): ‘Log-transformed sum of cadmium, lead and zinc (Met)’: explained variation = 11.7%, pseudo-*F* = 5.7, *p* (adj.) = 0.003; ‘Conductivity (Cond)’: explained variation = 6.1%, pseudo-*F* = 3.1, *p* (adj.) = 0.020; ‘Oxygen concentration (Oxy)’: explained variation = 4.0%, pseudo-*F* = 2.1, *p* (adj.) = 0.012; ‘Log-transformed sum pesticides (Pest)’: explained variation = 3.0%, pseudo-*F* = 1.6, *p* (adj.) = 0.045; ‘pH’ explained variation = 2.9%, pseudo-*F* = 1.6, *p* (adj.) = 0.047; ‘Total volume of released sewage for three days prior to sampling time (Sew)’: explained variation = 2.9%, pseudo-*F* = 1.6, *p* (adj.) = 0.036. Abbreviations: *AmpBor*—*Amphinemura borealis*, *AmpSul*—*A. sulcicollis*, *AthBil*—*Athripsodes bilineatus*, *BaeAffSc*—*Baetis* aff. *scambus*, *BaeFus*—*B. fuscatus*, *ChaeCfVi*—*Chaetopteryx* cf. *villosa*, *CyrTri*—*Cyrnus trimaculatus*, *DruAnn*—*Drusus annulatus*, *HabLau*—*Habrophlebia lauta*, *HaleCfTe*—*Halesus* cf. *tesselatus*, *HydInf*—*Hydatophylax infumatus*, *HydAng*—*Hydropsyche angustipennis*, *HydInc*—*H. incognita*, *HydInc/Pel*—*H. incognita*/*pellucidula*, *HydPel*—*H. pellucidula*, *HydSax*—*H. saxonica*, *HydSil*—*H. siltalai*, *LeuAlb*—*Leuctra albida*, *LeucCfAr*—*L.* cf. *armata*, *LeuBra*—*L. braueri*, *LeucCfPs*—*L.* cf. *pseudocingulata*, *LeucSp*—*Leuctra* sp., *LeuFus*—*L. fusca*, *LeuGen*—*L. geniculata*, *LeuIne*—*L. inermis*, *LeuNig—L. nigra*, *NemoCfUn*—*Nemoura* cf. *uncinata*, *NemoSp*—*Nemoura* sp., *NeuBim*—*Neureclipsis bimaculata*, *OdoAlb—Odontocerum albicorne*, *PhiLud*—*Philopotamus ludificatus*, *PolFla*—*Polycentropus flavomaculatus*, *PotaCfCi*—*Potamophylax* cf. *cingulatus*, *PotAffRot*—*P.* aff. *rotundipennis*, *PotRot*—*P. rotundipennis*, *ProHra*—*Protonemura hrabei*, *ProMey*—*P. meyeri/nitida*, *ProNit*—*P. nitida*, *ProtCfLa*—*P.* cf. *lateralis*, *PsyPus*—*Psychomyia pusilla*, *RhiSem*—*Rhithrogena semicolorata*, *RhitCfPu*—*R.* cf. *puytoraci*, *RhyaCfAu*—*Rhyacophila* cf. *aurata*, *RhyAffDo*—*R.* aff. *dorsalis*, *RhyAffNu*—*R.* aff. *nubila*, S*erFla/Per*—*Sericostoma flavicorne*/*personatum*, *SerIgn*—*Serratella ignita*.

**Figure 4 biology-11-00648-f004:**
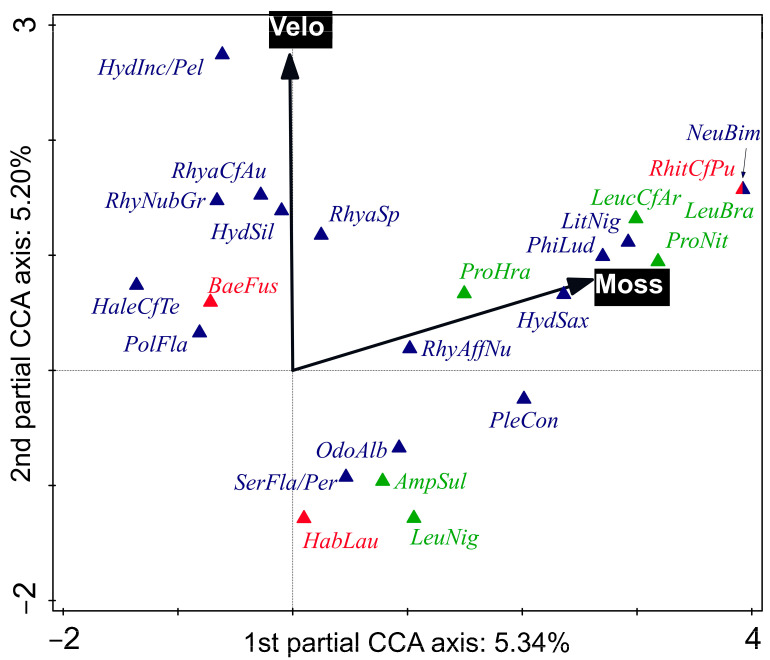
Species + subplot environmental variables (EVs) biplot: the first 24 best-fitting species are shown (Ephemeroptera, Plecoptera, and Trichoptera labelled in red, green, and blue, respectively). A partial Canonical correspondence analysis (CCA) was used. Ordination axes were constrained by the EVs selected by forward selection, which accounted for 10.54% of the explained variation. Test on first axis: pseudo-*F* = 0.1, *p* = 0.024; test on all axes: pseudo-*F* = 1.4, *p* = 0.001. Conditional effects (from the greatest to the lowest explained variation): ‘Arcsine-transformed proportion of mosses in choriotope (Mosses)’: explained variation = 5.3%, pseudo-*F* = 2.4, *p* (adj.) = 0.015 and ‘Mean current velocity (Velo)’: explained variation = 5.2%, pseudo-*F* = 2.5, *p* (adj.) = 0.023. Abbreviations: *AmpSul*—*Amphinemura sulcicollis*, *BaeFus*—*B. fuscatus*, *HabLau*—*Habrophlebia lauta*, *HaleCfTe*—*Halesus* cf. *tesselatus*, *HydInc/Pel*—*Hydropsyche incognita*/*pellucidula*, *HydSax—H. saxonica*, *HydSil*—*H. siltalai*, *LeuBra*—*L. braueri*, *LeucCfAr*—*L.* cf. *armata*, *LeuNig*—*L. nigra*, *LitNig*—*Lithax niger*, *NeuBim*—*Neureclipsis bimaculata*, *OdoAlb—Odontocerum albicorne*, *PhiLud*—*Philopotamus ludificatus*, *PleCon*—*Plectrocnemia conspersa*, *PolFla*—*Polycentropus flavomaculatus*, *ProHra*—*Protonemura hrabei*, *RhitCfPu*—*Rhithrogena* cf. *puytoraci*, *RhyaCfAu*—*Rhyacophila* cf. *aurata*, *RhyAffNu*—*R.* aff. *nubila*, *RhyaSp*—*Rhyacophila* sp. (early instar larvae), *RhyNubGr*—*R. nubila* gr., S*erFla/Per*—*Sericostoma flavicorne*/*personatum*.

**Figure 5 biology-11-00648-f005:**
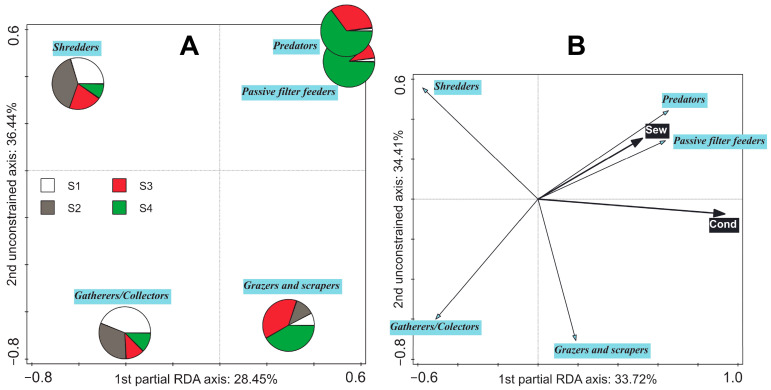
(**A**) Averaged feeding strategies pie chart and (**B**) Averaged feeding strategies + whole-plot environmental variables (EVs) biplot. Partial Redundancy analyses (RDA) were used. First ordination axes were constrained: (**A**) by the factor ‘Site’ that accounted for 30.98% of the explained variation and (**B**) by the EVs selected by forward selection, which accounted for 38.91% of the explained variation. (**A**) Test on first axis: pseudo-*F* = 5.4, *p* = 0.003; test on all axes: pseudo-*F* = 6.1, *p* = 0.026; (**B**) test on first axis: pseudo-*F* = 2.5, *p* = 0.006; test on all axes: pseudo-*F* = 5.7, *p* = 0.009. Conditional effects (from the greatest to the lowest explained variation): (**A**) ‘Site 4 (S4)’ explained variation = 22.0%, pseudo-*F* = 12.1, *p* (adj.) = 0.027; ‘Site 3 (S3)’ explained variation = 8.6%, pseudo-*F* = 5.2, *p* (adj.) = 0.090; and ‘Site 2 (S2)’ explained variation = 0.3%, pseudo-*F* = 0.2, *p* (adj.) = 0.943. (**B**) ‘Conductivity (Cond)’: explained variation = 32.5%, pseudo-*F* = 20.7, *p* (adj.) = 0.005 and ‘Total volume of released sewage for three days prior to sampling time (Sew)’: explained variation = 6.4%, pseudo-*F* = 4.4, *p* (adj.) = 0.045.

**Figure 6 biology-11-00648-f006:**
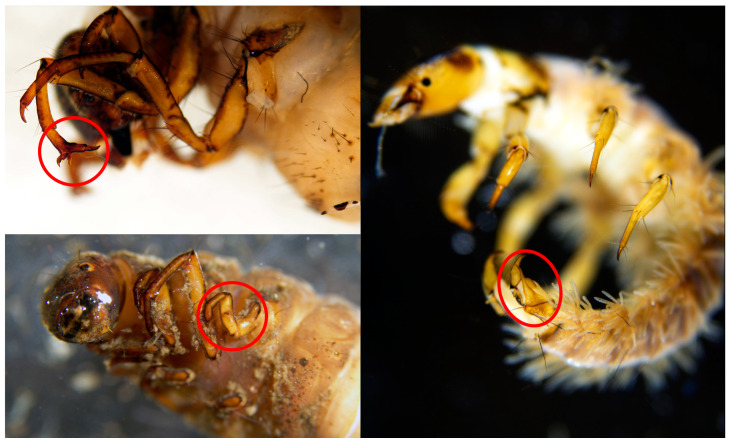
Malformed limbs of *Halesus* cf. *tesselatus* (two pictures on the left) and *Rhyacophila* cf. *aurata* (right side) sampled at Sites 3 and 4.

**Table 1 biology-11-00648-t001:** Resulted mean physicochemical parameters measured at the given sites and concentrations of metals, pesticides, and active pharmaceutical compounds (PhACs) in water samples taken at these sites.

	Site 1	Site 2	Site 3	Site 4
	n = 4 Mean ± SEM	n = 4 Mean ± SEM	n = 4 Mean ± SEM	n = 4 Mean ± SEM
**Temperature** (**°C**)	12.05 ± 0.34	13.33 ± 0.61	17.05 ± 1.11	16.80 ± 0.68
**pH**	7.09 ± 0.24	7.35 ± 0.17	7.45 ± 0.17	7.45 ± 0.09
**Oxygen concentration** (**mg·L^–1^**)	9.90 ± 0.10	9.02 ± 0.18	8.91 ± 0.16	7.61 ± 0.27
**Conductivity** (**µS·cm^–1^**)	71.40 ± 5.80	84.35 ± 5.98	390.60 ± 39.18	571.00 ± 56.74
**Concentrations in Water Samples**				
**Cadmium** (**µg·L^–1^**)	0.26	0.23	7.90	3.80
**Lead** (**µg·L^–1^**)	5.80	9.80	64.00	53.00
**Zinc** (**µg·L^–1^**)	15.00	13.00	1200.00	530.00
**Sum pesticides** (**ng·L^–1^**)	132.00	70.32	462.90	877.87 ± 125.71 (n = 3)
**Sum PhACs** (**ng·L^–1^**)	15.17	13.40	874.58	4636.73 ± 254.79 (n = 3)

**Table 2 biology-11-00648-t002:** Monthly volumes of untreated or ‘poorly treated’ wastewater released from the wastewater treatment plant into Site 4 in 2019 and 2020.

	Volume [m^3^]	
	2019	2020
January	164,799	1522
February	80,750	55,224
March	58,862	35,861
April	183	3239
May	26,930	24,749
June	34,483	130,016
July	13,340	11,305
August	26,333	22,382
September	817	12,064
October	4967	67,559
November	3270	5941
December	71	299
**Total**	**414,805**	**370,161**

**Table 3 biology-11-00648-t003:** Resulted mean current velocities measured for the macrozoobenthic samples from each site.

	Site 1	Site 2	Site 3	Site 4
Velocity (m·s^–1^)	n = 12 Mean ± SEM	n = 12 Mean ± SEM	n = 12 Mean ± SEM	n = 9 Mean ± SEM
**Average**	0.19 ± 0.04	0.14 ± 0.03	0.27 ± 0.04	0.29 ± 0.05
**Maximum**	0.44 ± 0.09	0.32 ± 0.06	0.70 ± 0.14	0.62 ± 0.10
**Minimum**	0.04 ± 0.04	0.02 ± 0.03	0.04 ± 0.04	0.06 ± 0.05

**Table 4 biology-11-00648-t004:** Resulted mean choriotopic percentual compositions empirically estimated for each macrozoobenthic sample at all given sites.

	Site 1	Site 2	Site 3	Site 4
Choriotopic Parameter	n = 12 Mean ± SEM (%)	n = 12 Mean ± SEM (%)	n = 12 Mean ± SEM (%)	n = 12 Mean ± SEM (%)
**Megalithal** Large cobbles, boulders and blocks, bedrock; >40 cm	5.75 ± 2.63	18.29 ± 7.87	6.08 ± 2.16	18.50 ± 7.30
**Macrolithal** Coarse blocks, cobbles, gravel and sand; 20–40 cm	29.92 ± 6.88	17.38 ± 5.28	19.10 ± 4.62	12.60 ± 2.95
**Mesolithal** Fist to hand-sized cobbles; 6.3–20.0 cm	16.58 ± 3.75	22.29 ± 3.37	28.50 ± 2.41	19.70 ± 3.48
**Microlithal** Coarse gravel; 2.0–6.3 cm	11.08 ± 1.89	8.67 ± 2.23	18.30 ± 3.29	11.20 ± 0.99
**Akal** Fine to medium-sized gravel; 0.2–2.0 cm	15.50 ± 3.32	5.21 ± 0.93	11.60 ± 1.97	11.90 ± 2.06
**Psammal** Sand; <0.2 cm	3.88 ± 1.12	1.21 ± 0.58	3.00 ± 1.04	4.50 ± 2.00
**Xylal** Deadwood, cones	4.79 ± 1.33	10.38 ± 2.71	2.04 ± 0.82	1.63 ± 0.80
**Coarse particulate organic matter** CPOM; coarse deposits, e.g., leaves	1.83 ± 0.82	2.04 ± 0.81	0.88 ± 0.26	3.42 ± 1.38
**Fine particulate organic matter** FPOM; fine deposits	0.75 ± 0.41	3.96 ± 1.54	1.25 ± 0.62	4.25 ± 1.53
**Debris** Hard and coarse matter—organic or inorganic	2.67 ± 0.77	6.63 ± 2.04	3.13 ± 0.57	4.63 ± 1.10
**Mosses** *Fontinalis antipyretica* and Marchantiophyta at the Site 1	6.25 ± 3.27	2.71 ± 1.34	1.58 ± 1.21	2.13 ± 0.83
**Filamentous algae**	0	0	0	2.33 ± 1.37
**Roots of riparian trees** *Alnus* sp., *Salix* sp. or conifers	0.58 ± 0.42	1.25 ± 0.86	3.33 ± 1.70	0
**Submerged riparian vegetation**	0.42 ± 0.40	0	1.25 ± 0.63	0.83 ± 0.54
**‘****Wet wipes’** Anthropogenic trash	0	0	0	1.38 ± 0.61
**‘Metal sheets’** Anthropogenic trash	0	0	0	1.17 ± 1.12

**Table 5 biology-11-00648-t005:** Results of testing the significance of differences within factors site and sampling time (both four-level factors) in given response variables using Generalised linear models (GLMs) or Generalised linear mixed-effects models (GLMMs). Significant (*p* < 0.05) results are in bold text. No significant interaction between site and sampling time was detected for any response variable (*p* > 0.05). E—Ephemeroptera, P—Plecoptera, and T—Trichoptera.

	Factors	
Response Variable	Site	Sampling Time
**EPT abundance**	*χ*^2^_3_ = 1.19, *p* = 0.76	*χ*^2^_3_ = 3.96, *p* = 0.27
**EPT richness**	***F*_3,44_ = 2.82, *p* = 0.049**	*F*_3,41_ = 1.14, *p* = 0.34
**E abundance**	*χ*^2^_3_ = 1.22, *p* = 0.75	*χ*^2^_3_ = 7.56, *p* = 0.055
**E richness**	*F*_3,44_ = 0.14, *p* = 0.93	***F*_3,44_ = 3.19, *p* = 0.03**
**P abundance**	***χ*^2^_3_ = 56.33, *p* < 0.001**	***χ*^2^_3_ = 9.70, *p* = 0.02**
**P richness**	***χ*^2^_3_ = 124.03, *p* < 0.001**	***χ*^2^_3_ = 8.14, *p* = 0.04**
**T abundance**	***χ*^2^_3_ = 12.59, *p* = 0.006**	*χ*^2^_3_ = 6.11, *p* = 0.11
**T richness**	*F*_3,44_ = 1.03, *p* = 0.39	*F*_3,44_ = 0.43, *p* = 0.73
**E family richness**	***F*_3,41_ = 6.83, *p* < 0.001**	*F*_3,41_ = 2.48, *p* = 0.074
**P family richness**	***F*_3,44_ = 23.70, *p* < 0.001**	*F*_3,41_ = 1.02, *p* = 0.39
**T family richness**	*F*_3,44_ = 0.47, *p* = 0.70	*F*_3,44_ = 0.23, *p* = 0.88

## Data Availability

Not applicable.

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
