# Peer review of "Effects of Trace Metals and Municipal Wastewater on the Ephemeroptera, Plecoptera, and Trichoptera of a Stream Community"

_biology, 2022, doi:10.3390/biology11050648_

Round 1

Reviewer 1 Report

These are my main comments on the manuscript (Biology-1695692) entitled “Effects of trace metals and municipal wastewater on the Ephemeroptera, Plecoptera and Trichoptera of a stream community”. The manuscript “We sampled a stream ecosystem affected by mining and smelting industries and communal wastewaters with pronounced concentrations of cadmium lead and zinc, as well as high levels of pesticides, pharmaceuticals, illegal drugs and sewage-derived organic matter. The results showed how these insect communities react to human-induced gradients in natural environments. Following moderated revisions should be incorporated in the manuscript prior to acceptance.

L.12: Delete “by research institutions”

L.21: Delete “significantly”

Ls.41-42: Keywords should be in alphabetic order. Also, keywords serve to widen the opportunity to be retrieved from a database. To put words that already are into title and abstracts makes KW not useful. Please choose terms that are neither in the title nor in abstract.

L.46: … stoneflies (Plecoptera), and…

Ls.48 and 52: Two “particularly” is not nice

L.49: … deposits are low and essential…

L.56: …threats are objectively…

L.59: … habitat structure, and…

L.88: … phthalates, and trace…

L:197: … whole plot was…

L.219: … Sites 1, 2, and 3…

L.236: … Buffagni et al. [37], and…

L.272-274: Rephrase these sentences

L.280: Deleter “approximately”

L.300: … and debris, occurrence of filamentous algae and…

L.302: … of metals, pesticides, and…

Table 1, 2, and 4: What statistical method did you use for data analysis?

L.313: … 29 stoneflies, and 47…

L.319: Change “was” by “were”

L.341: … L. fusca, and…

L.345: … Amphinemura, and Nemoura…

L.352: … Sites 2, 3, and 4…

L.358: … Halesus, and Chaetopteryx…

L.361: … H. angustipennis, and H…

L.472: … a total of 20 specimens…

L.517: … at Sites 3 and 4…

L.528: … found in our study…

L.531: … of cadmium, copper, and zinc…

L.681: … polycentropodids, and rhyacophilids…

Author Response

Dear reviewer,

We are submitting our revised manuscript "Effects of trace metals and municipal wastewater on the Ephemeroptera, Plecoptera and Trichoptera of a stream community".

In this document below, please find the point-by-point answers to all constructive comments and revisions. We tried to fulfil all requirements and suggestions to improve the manuscript and increase its scientific quality. We respond to all suggestions and requirements. Concurrently, we added our own improvements (page n. 6 of this document).

Please, find the manuscript and supplementary materials with modifications visible by track changes (inside a zip folder “Let et al., 2022 revised”: word files – “Manuscript_M._Let_ revised” and “Supplementary_materials_M._Let_revised”, respectively).

Thank you for your time and effort when evaluating our revised contribution.

Yours faithfully,

Marek Let (on behalf of collective authors)

Response to Comments:

L.12: Delete "by research institutions"

L.21: Delete "significantly"

Our response: we deleted both.

Ls.41-42: Keywords should be in alphabetic order. Also, keywords serve to widen the opportunity to be retrieved from a database. To put words that already are into title and abstracts makes KW not useful. Please choose terms that are neither in the title nor in abstract.

Our response: We changed keywords that are not neither in the tittle, nor in the abstract, and we used alphabetic order. Keywords are listed now as follows: Anthropogenic disturbances; Aquatic insect; Environmental gradients; Heavy metals; Industrial pollution; Wastewater treatment plant.

L.46: … stoneflies (Plecoptera), and…

Our response: we added the comma.

Ls.48 and 52: Two "particularly" is not nice

Our response: We rewrote the second "particularly" to "namely".

L.49: … deposits are low and essential…

Our response: We removed the comma.

L.56: …threats are objectively…

Our response: We replaced be for are.

L.59: … habitat structure, and…

Our response: we added the comma.

L.88: … phthalates, and trace…

Our response: we added the comma.

L.197: … whole plot was…

Our response: We replaced plural for singular.

L.219: … Sites 1, 2, and 3…

Our response: We added comma the comma.

L.236: … Buffagni et al. [37], and…

Our response: We added the comma.

L.272-274: Rephrase these sentences

Our response: We rephrased as follows:

Several environmental gradients including increasing temperature, pH, and conductivity and decreasing oxygen concentrations changed along with the longitudinal downstream profile.

L.280: Delete “approximately”

Our response: We deleted it.

L.300: … and debris, occurrence of filamentous algae and…

Our response: We added the comma.

L.302: … of metals, pesticides, and…

Our response: We added the comma.

Table 1, 2, and 4: What statistical method did you use for data analysis?

The results in tables 1, 2, and 4 are mostly descriptive statistics (arithmetic means ± Standard Error of Mean; SEM). We decided to not test the significance between sites in basic physicochemical parameters (Table 1) because we had only four replicates per one site and, in addition, there was a time dependence within each set of four sites. In the case of contamination (Table 1), it was neither possible because we estimated the contamination at all sites at one time. In the case of choriotopes (Table 4), we made RDA and non-parametric permutation test but differences between sites were marginal (see Supplementary materials – Figure S4). We did not test individual choriotopic parameters separately by univariate statistical tests. In Table 4, we present descriptive statistics same as in Tables 1 and 2. But it should be mentioned that the compositions of choriotopes were estimated empirically. In contrast, the results in Tables 1 and 2 are measured using devices and standardised analytical methods (see Supplementary materials – Materials and Methods). Since main output of these study was analyses of the EPT community response, we believe that summarizing results of environmental data in this way (descriptive statistics) is a standard option and it is understandable for readers. Nevertheless, we slightly adjusted the tittles of these tables (see Our improvements, page 6 of this document).

L.313: … 29 stoneflies, and 47…

Our response: We added the comma.

L.319: Change “was” by “were”

Our response: We corrected accordingly.

L.341: … L. fusca, and…

Our response: We added the comma.

L.345: … Amphinemura, and Nemoura…

Our response: We added the comma.

L.352: … Sites 2, 3, and 4…

Our response: We added the comma.

L.358: … Halesus, and Chaetopteryx…

Our response: We added the comma.

L.361: … H. angustipennis, and H…

Our response: We added the comma.

L.472: … a total of 20 specimens…

Our response: We added “of”.

L.517: … at Sites 3 and 4…

Our response: We corrected accordingly.

L.528: … found in our study…

Our response: We corrected accordingly.

L.531: … of cadmium, copper, and zinc…

Our response: We corrected accordingly.

L.681: … polycentropodids, and rhyacophilids…

Our response: We added the comma.

Our improvements:

L.303–304, L. 309, L. 312–313: We changed the tittle of tables.

  1. 460: We removed redundant space.
  2. 538: We added comma.

Supplementary materials

L.62: The name of used scales was wrong. We corrected it.

Reviewer 2 Report

The work of M. Let et al. titled “Effects of trace metals and municipal wastewater on the Ephemeroptera, Plecoptera and Trichoptera of a stream community” covers the impact of mayflies (Ephemeroptera), stoneflies (Plecoptera) and caddisflies (Trichoptera) (EPT) on the trace metal pollution of water bodies. This study attempts to shed light how EPT community interact with human societies and their industrial footprint and the potential use as water quality bioindicator. The study shows robust statistical analysis of those parameters which affect more the water quality. The achieved results are well-discussed during the main body of the reported manuscript. The gathered knowledge may aid to create alternative methodologies and protocols to control the water quality which could contribute to improve the mankind welfare. The scientific paper is well written. In my opinion the present manuscript is innovative and the methodological approached used matches with the scope of Biology. For the above described reasons, I recommend the publication in Biology once the following minor remarks will be fixed:

--------

INTRODUCTION

The manuscript already cites the detrimental impact of predators in EPT populations (overall the Fig. S7 and Fig. S8 in Supplementary information) but it lacks a quantification of their activity in other studies. Some information should be added on this regard with the corresponding reference citations.

--------

RESULTS

Result section is well-structured and clearly explained. However, authors should take care about the following aspects:

  1. Table 1 lacks the standard error of the mean related to trace metals and organic pesticides. Please, introduce this data.

  1. I consider it would convenient to introduce an additional Figure indicating the taxa richness and percentage of EPTs by metal index for the 4 selected studied river regions. A bibliography reference should be also added indicating previous studies where this kind of approach is done (e.g. Giddings, E.M.P.; Homberger, M.I.; Hadley, H.K. Trace-metal concentrations in sediment and water and health of aquatic macroinvertebrate communities of streams near Park City, Summit Country, Utah. https://doi.org/10.3133/wri014213. If the authors consider more opportune, box plots could also be done instead the aforementioned suggestion. These additions will facilitate the most relevant findings comprehension by the potential readers.

Author Response

Dear reviewer,

We are submitting our revised manuscript "Effects of trace metals and municipal wastewater on the Ephemeroptera, Plecoptera and Trichoptera of a stream community".

In this document below, please find the point-by-point answers to all constructive comments and revisions. We tried to fulfill all requirements and suggestions to improve the manuscript and increase its scientific quality. We respond to all suggestions and requirements. Concurrently, we added our own improvements (page n. 6 of this document).

Please, find the manuscript and supplementary materials with modifications visible by track changes (inside a zip folder “Let et al., 2022 revised”: word files – “Manuscript_M._Let_ revised” and “Supplementary_materials_M._Let_revised”, respectively).

Thank you for your time and effort when evaluating our revised contribution.

Yours faithfully,

Marek Let (on behalf of collective authors)

Response to Comments:

INTRODUCTION

The manuscript already cites the detrimental impact of predators in EPT populations (overall the Fig. S7 and Fig. S8 in Supplementary information) but it lacks a quantification of their activity in other studies. Some information should be added on this regard with the corresponding reference citations.

Our response: According to published information, we do not believe that hydropsychids and polycentropodids (the most abundant EPT predators) would have a causal detrimental impact on EPT population. Nevertheless, we cannot reject it. We added this hypothesis to the discussion and we cited several studies connected to this topic (Lines 618–624). According to Figures S7 and S8, it looks like predators and passive filter feeders (hydropsychids and polycentropodids) could compete out the other feeding strategies but this can be also caused by a high amount of drifting sewage derived particulate organic matter which could represent the main food source for these facultative predators. Thus net-spinning (passive filtration) may be logically the most advantageous strategy. This issue represents just a disadvantage of the application of species traits from a database because there can be plasticity in food utilization within individual species or related species group and between sites and times that cannot be reflected using this approach. It should be pointed out that Figure S8 (15-d Sewage) is based only on marginally significant relationships (p > 0.05) tested within RDA, although the individual visualised responses of feeding strategies along the explanatory variable were significant. Only the conductivity and the 3-d Sewage (Figure S7) had the significant power to explain the composition feeding strategy. We interpret the trend in feeding strategies with the highest 15-d sewage (Fig. S8) rather as the reaction of the community to high discharge (insect drift) combined to 15-d sewage volumes. The samples from the site affected by municipal wastewaters were more similar to ones from the upstream site non-affected by wastewaters, in spite of the highest 15-d volumes (see Discussion and Figure S5A, where these samples caused an overlay of S3 and S4). The formal EPT community was likely impaired by high discharge combined with sewage but the sewage inflow stopped more than three days before our sampling in June and taxa relatively sensitive to organic pollution colonised the stretch. Thus, we add it to supplementary materials as an interesting but hardly clear trend.

--------

RESULTS

Result section is well-structured and clearly explained. However, authors should take care about the following aspects:

  1. Table 1 lacks the standard error of the mean related to trace metals and organic pesticides. Please, introduce this data.

Our response: We cannot add standard errors where we did not sample repeatedly. There was usually one sample, except for Site 4, where we sampled at 3 sites along with a longitudinal downstream profile. This is explicitly written in Materials and Methods (Lines 214–217):

(ii) pesticide and PhAC concentrations in water (measured from samples taken on 11 August 2020), (iii) cadmium, lead and zinc concentrations measured in sediments and water (measured from samples taken on 24 September 2020)…

We must admit that this was not ideal; however, we had also monthly data from the Povodí Vltavy, State Enterprise, and the Czech Hydrometeorological Institute (national water management institutions in the Czech Republic). When we analysed their data from the year of samplings as well as other years, we encountered several complications, and we decided rather use our measurements, even though there were less replications because it best reflected differences between localities (only Site 4 and Aux. site 4 were analysed for a wide range of compounds by the given water management institutions and analyses of many contaminants at the Site 4 were stopped just from the beginning of 2020). On the other hand, values about trace metal contamination were still available, particularly concentration in water. They very well corresponded to the values measured in our samples (it was analysed in the same laboratory because our home institution does not analyse this type of contamination). Unfortunately, the most contaminated Site 3 has not been regularly monitored by water management for several decades may be from political reasons (the active industrial complex was privatised in the 90ies). On the other hand, the contamination of the Litavka river by heavy metals at sites of our study has been under the scope of many researchers, e.g.:

Žák, K., Rohovec, J., & Navrátil, T. (2009). Fluxes of heavy metals from a highly polluted watershed during flood events: a case study of the Litavka River, Czech Republic. Water, Air, and Soil Pollution, 203(1), 343-358.

Friedlova, M. (2010). The influence of heavy metals on soil biological and chemical properties. Soil and Water Research, 5(1), 21-27.

Matys Grygar, T., Faměra, M., Hošek, M., Elznicová, J., Rohovec, J., Matoušková, Š., & Navrátil, T. (2021). Uptake of Cd, Pb, U, and Zn by plants in floodplain pollution hotspots contributes to secondary contamination. Environmental Science and Pollution Research, 28(37), 51183-51198.

Thus, we are convinced there is no doubt about the relevance of our result from the perspective of contamination.

  1. I consider it would convenient to introduce an additional Figure indicating the taxa richness and percentage of EPTs by metal index for the 4 selected studied river regions. A bibliography reference should be also added indicating previous studies where this kind of approach is done (e.g. Giddings, E.M.P.; Homberger, M.I.; Hadley, H.K. Trace-metal concentrations in sediment and water and health of aquatic macroinvertebrate communities of streams near Park City, Summit Country, Utah. https://doi.org/10.3133/wri014213. If the authors consider more opportune, box plots could also be done instead the aforementioned suggestion. These additions will facilitate the most relevant findings comprehension by the potential readers.

We tried to use the metal index according to Giddings et al. (2001). Since we estimated the total abundance of macrozoobenthos (all taxa, not only EPT) in 19 samples from a total of 48, we used relative abundance to model the trend between the Metal index (calculated from water concentrations because the sediment concentrations had variable content of organic carbon; see Supplementary materials – Table S3) and relative abundance of EPT taxa. We used GLM without implementing a random effect of sampling time with quasibinomial distribution and overdispersion was massive. The model was not significant: F1,17 = 0.05, p = 0.83 (the graph below):

We have not established total abundance in the remaining samples. The establishment of species richness is also out of our capabilities (because abundant chironomids are a very complicated group with time-consuming determination process). For this reason, we aimed at only EPT because morphological determination is less difficult, its pollution sensitivity, and there are more comparable articles.

Nevertheless, we tried to model the trend between Metal index and EPT richness that was not significant (F1,46 = 2.08, p = 0.16), the trend with EPT abundance as response (F1,46 = 0.02, p = 0.88), and the trend with EPT family richness as response that was significant (χ21 = 13.42, p < 0.001):

We added this improvement into the Supplementary materials (Lines 69–93); we created new chapter results and add the significant relationship as Figure S9.

Our improvements:

L.303–304, L. 309, L. 312–313: We changed the tittle of tables.

  1. 460: We removed redundant space.
  2. 538: We added comma.

Supplementary material

L.62: The name of used scales was wrong. We corrected it.
